# Personality Disorders and Traits of ABC Clusters in Fibromyalgia in a Neurological Setting

**DOI:** 10.3390/biomedicines11123162

**Published:** 2023-11-28

**Authors:** Dmitry V. Romanov, Tatiana I. Nasonova, Aleksey I. Isaikin, Olga V. Filileeva, Andrey M. Sheyanov, Polina G. Iuzbashian, Evgenia I. Voronova, Vladimir A. Parfenov

**Affiliations:** 1Department of Psychiatry and Psychosomatics, I.M. Sechenov First Moscow State Medical University (Sechenov University), 119435 Moscow, Russia; olga.filileeva@gmail.com (O.V.F.); andrey.sheyanov.00@mail.ru (A.M.S.); astartes@ya.ru (P.G.I.); voronova_e@mail.ru (E.I.V.); 2Mental Health Research Center, 115522 Moscow, Russia; 3International Institute of Psychosomatic Health, 127550 Moscow, Russia; 4Department of Nervous Diseases and Neurosurgery, I.M. Sechenov First Moscow State Medical University (Sechenov University), 119435 Moscow, Russia; tatiana.martova@yandex.ru (T.I.N.); alexisa68@mail.ru (A.I.I.); vladimirparfenov@mail.ru (V.A.P.); 5Institute of Clinical Medicine, I.M. Sechenov First Moscow State Medical University (Sechenov University), 119435 Moscow, Russia; andrey.sheyanov.00@mail.ru

**Keywords:** fibromyalgia, personality disorders, Axis II, personality disorder clusters, personality traits, borderline, schizotypal, schizoid, avoidant

## Abstract

Background: Evidence suggests that there is substantial comorbidity between fibromyalgia and Axis II pathology (i.e., personality disorders—PDs). The aim of the current study was to find out the exact cluster (A, B, C) of PDs or traits that are more prominent in FM and may be predictors of FM diagnosis. Methods: Data from 86 subjects (53 with FM and 33 controls without FM) were analyzed in an observational, cross-sectional, comparative study in a neurological setting. The assessment of categorical PDs and traits was performed independently with the Structured Clinical Interview for Personality Disorders (SCID-II). Binary logistic regression was used to determine FM predictors among PD traits. Results: Compared with controls, FM patients had a higher rate of PD diagnoses (56.7 vs. 18.2%, *p* < 0.001). However, the rate was significantly higher only for borderline PD diagnosis (28.3% vs. 6.1% *p* < 0.05). The binary logistic regression analysis showed that schizotypal and schizoid (cluster A), borderline (cluster B), and dependent (cluster C) personality traits may be significant predictors of fibromyalgia (Nagelkerke R^2^ = 0.415). Conclusions: Our results may reflect the association of FM with personality traits of all three PD clusters: A (eccentric), B (dramatic), and C (anxious). However, the most consistent evidence seems to be for borderline PD.

## 1. Introduction

Fibromyalgia (FM) is a chronic pain syndrome that is characterized by persistent musculoskeletal pain, fatigue, sleep disturbances, and functional symptoms [1,2]. It is believed to be a dysfunction of the CNS, but no definite structural lesion has been identified so far [3]. FM syndrome has recently been defined in the UK clinical guidelines by the Royal College of Physicians as a multifactorial disorder with neurophysiological, immunological, and cognitive elements [4]. It is also stressed that although the precise cause of FM remains unknown, the central issue is proposed to be abnormal pain processing within the nervous system. However, the idea of fibromyalgia as a primary pain disorder with a neurobiological basis contends with fibromyalgia as a broader biopsychosocial disorder [5]. Thus, to avoid dichotomous neurological/immunological/rheumatological vs. psychological/psychiatric/psychosomatic conceptualization of the disorder, there is a focus on the need to adopt a biopsychosocial perspective for understanding and addressing patients with FM, noting that biological, psychosocial, and behavioral factors function interdependently to affect a person’s experience and adaptation [6].

Regardless of the theoretical controversies, FM is peculiar for its high comorbidity with mental disorders [7]. Among the most frequent comorbidities are mood disorders, anxiety, somatoform, obsessive-compulsive, and personality disorders (PDs). PDs are of particular interest and remain relatively underinvestigated in FM.

A personality disorder (PD) is defined as an enduring pattern of inner experience and behavior that deviates markedly from the norms and expectations of the individual’s culture, is pervasive and inflexible, has an onset in adolescence or early adulthood, is stable over time, and leads to distress or impairment [8].

PDs are rather peculiar clinical entities and differ from other psychiatric conditions in a set of features. For instance, in DSM-IV, PDs are placed among Axis II disorders in contrast to Axis I disorders (affective, anxious, obsessive-compulsive, etc.). There is strong evidence of high Axis I and Axis II disorder comorbidity [9], as well as high comorbidity of PDs with other medical and biopsychosocial conditions [10,11] and comorbidity within subtypes of PDs and between PD traits [12,13].

PD onset often precedes the onset of Axis I disorders, and PDs are considered by some authors as predisposing conditions and vulnerability factors or predictors for other psychopathology and even for some somatic or functional conditions [14]. Another important issue in PD research is the existence of two basic approaches to PD conceptualization: categorical (e.g., in DSM-III-R, DSM-IV, and ICD-10 [15,16,17]) and dimensional (e.g., the “Bif Five” model and ICD-11 [18,19]). This makes it difficult to compare the data from individual studies, although each of the approaches has its advantages. However, there is an approach that combines both categorical and dimensional paradigms (e.g., in DSM-5-TR, categorical PDs coexist with the alternative Model for Personality Disorders, i.e., “hybrid” model [8]). We believe that the latter is highly promising in a scientific perspective.

Most of the existing studies show that the proportion of PDs diagnosed in patients with FM appears far greater than that found in the general population [20,21,22]. However, data about subtypes of PDs related to FM remain controversial. For instance, there is evidence of “dramatic” cluster B PD predominance (e.g., borderline and histrionic) [7,21]. The contrasting data suggest that “anxious” cluster C PDs (e.g., avoidant and obsessive-compulsive) are the most common in FM [23,24,25,26]. In addition, there are studies that report high rates of both cluster B and C PDs [27]. However, no data exist regarding the impact of cluster A PDs, although there are some reports about FM’s co-occurrence with paranoid and schizoid PDs [23,25,28].

Currently known FM predictors include female sex, impaired sleep, few years of education, sleeping problems, overweight (BMI), rheumatoid and osteo-arthritis, other pain conditions (e.g., frequent headache, persistent back and neck pain, migraine), and psychological factors (i.e., alexithymia and psychological distress) [29,30,31]. However, very little is known about the personality predictors of FM. Among these are neuroticism and extraversion [32,33,34]. However, categorically based PDs and/or PD traits are still underinvested in this regard.

Thus, the aim of the current study was two-fold: (1) to determine the exact PDs and traits among A, B, and C clusters that are more frequent and/or severe in FM and (2) to establish PD traits that may be predictors of FM diagnosis.

## 2. Materials and Methods

An observational, cross-sectional, comparative study was conducted at Kozhevnikov Neurology Clinic of Sechenov University in Moscow, Russia, between January 2020 and December 2022.

The study sample consisted of a main and a control group. The main group included 53 patients with FM (47 female, mean age 46.8 ± 14.6 years). The control group comprised 33 subjects without FM (24 female, mean age 43.6 ± 12.4 years). The study groups did not differ significantly in mean age, sex distribution, and marital status. The frequency of unemployed participants was significantly higher in FM patients (*p* < 0.001) as a sign of professional disability due to poor health status. This observation was also supported with the health-related quality-of-life measure EQ-5D (EuroQol 5 Dimensions Health-related Quality of Life Questionnaire [35]). The EQ-5D-derived 100-point visual analog scale (VAS) showed significantly lower self-assessed health in FM (46.8 vs. 82.2, *p* < 0.001). FM patients also had a significantly higher rate of an official disability (20.8 vs. 3.0%, *p* = 0.021), a lower educational status (54.7% vs. 93.9%, *p* = 0.01), and a higher mean BMI (27.0 vs. 23.4, *p* = 0.008). Fewer years of education and higher BMI (body mass index) are among the established predictors of FM [29]. These findings support the notion that FM is a highly debilitating condition. Mean scores of self-questionnaires for FM (the Revised Fibromyalgia Impact Questionnaire (FIQR) [36] and the Fibromyalgia Rapid Screening Tool (FIRST) [37]) were predictably significantly higher in FM subjects than in healthy controls. Also, FM patients had significantly higher levels of self-measured anxiety and depression (the Hospital Anxiety and Depression Scale (HADS) [38]), as well as somatization (Screening for Somatoform Symptoms scale (SOMS-2) [39]). The sociodemographic, health status, and psychometric characteristics of the main and control groups of the study are summarized in Table 1.

Organic (non-functional) causes of pain were excluded due to extensive somatic and neurological examination (consultations with rheumatologists and neurologists). In cases where we doubted the nature of any presented pain symptoms, additional medical examinations were performed: laboratory examinations, magnetic resonance imaging (MRI), X-ray computed tomography of the spine, and electroneuromyography (ENMG). The examination was available due to the study site location at the large University Medical Centre hosting multi-field hospitals. In addition, patients with underlying medical conditions that could explain their complaints were excluded from this study.

The diagnosis of FM was based on the criteria of the American College of Rheumatology (ACR, 2016). The mean number of tender points in patients with FM was 11.3 ± 3.3. The mean duration of FM was 7 years and ranged from 2 to 15 years. The mean intensity of the pain syndrome (numeric rating scale, NRS) was 7.1 ± 1.9 points.

To classify the severity of fibromyalgia, the Fibromyalgia Impact Questionnaire (FIQR) was used, which uses a Likert-type scale and contains twenty-one items divided into three components (function, widespread impact, and severity of symptoms), and which considers a maximum of 100 points, classifying FM as mild (0–42 points), moderate (43–59 points), severe (60–74 points), and extreme (75–100 points) [36,40]. The mean FIQR score for FM subjects in our study was 54.1 ± 18.8 and within the range for moderate FM severity. However, 22 (40.0%) FM patients had FIQR scores > 59 points (severe FM).

To identify PDs in both the main group and the control group, all study participants were consulted by psychiatrists (D.V.R., O.V.F.). Written consent to the consultation with a psychiatrist was obtained from all participants in the study. The assessment of categorical personality disorders (Axis II) was performed with the Structured Clinical Interview for DSM-IV Personality Disorders (SCID-II) (SCID-II/PQ) [41]. Administration of the SCID-II requires a two-step approach [42]. First, subjects completed the 119-item SCID-II Personality Questionnaire (SCID-II/PQ), which uses a Yes/No response. Each of the questions corresponds to a diagnostic criterion (DP trait) for either one of the main text PDs or the two additional PDs listed in Appendix B of DSM–IV (i.e., Passive-Aggressive and Depressive PD). After questionnaire completion, the interviewer identified those PDs for which respondents endorsed sufficient criteria for PD diagnoses (threshold point scores/traits sufficient for a particular PD). Persons meeting self-report criteria for any given PD were then administered the corresponding portions of the SCID-II interview to assign a formal diagnosis. As there is strong evidence of high overlap between PD traits and a lack of pure prototypical cases, we used a “hybrid” approach. First, we diagnosed PDs according to the categorical paradigm (“one patient may have only one PD”) based on the most prominent personality pattern for each patient. Then, we analyzed PD trait sets peculiar for particular PDs as overlapping or comorbid entities (dimensions). Mean numbers of particular PD traits per study group and trait severity (number of traits) in individual patients were assessed. Certain PD traits were considered significant when PD trait sets exceeded the thresholds for SCID-II PDs: avoidant (AVPD ≥ 4 of 7 points), dependent (DPD ≥ 5 of 8 points), obsessive-compulsive (OCPD ≥ 4 of 9 points), passive-aggressive (PAPD ≥ 4 of 8 points), depressive (DRPD ≥ 5 of 8 points), paranoid (PRPD ≥ 4 of 8 points), schizotypal (STPD ≥ 5 of 11 points), schizoid (SCPD ≥ 4 of 6 points), histrionic (HIPD ≥ 5 of 7 points), narcissistic (≥5 of 17 points), borderline (BPD ≥ 5 of 15 points), and antisocial (ASPD ≥ 3 of 15 points).

Quantitative variables were checked for normality of distribution using the Shapiro–Wilk test. If the null hypothesis that data follow a normal distribution was rejected, a non-parametric Mann–Whitney U-test was used to compare groups. Pearson’s criterion χ2 was used to compare groups by qualitative variables. Binary logistic regression analysis was performed to elucidate FM predictors among SCID-II-derived personality traits. FM diagnosis was used as a dependent binary variable: positive diagnosis in the main group vs. negative diagnosis in the control group. As independent continuous quantitative variables, twelve SCID-II-derived PD trait mean numbers were used. The assumptions for binary logistic regression were fulfilled as the dependent variable was dichotomous, no outliers existed in the PD trait mean numbers, and there was no high correlation or multicollinearity among the PD traits as predictors. Statistical analysis was carried out using IBM SPSS Statistics for Windows, Version 27.0. IBM Corp, Armonk, NY, USA; 2020.

## 3. Results

The comparison between the main and control groups revealed that 56.7% of patients with FM fulfilled the criteria for a single PD (*n* = 30), and only 18.2% of control subjects had a PD diagnosis (*n* = 6), while most of the controls (*n* = 27, 81.8%) did not reach the diagnostic threshold for any PD (*p* < 0.001). Half of the PD-positive FM subjects were diagnosed with BPD (*n* = 15), followed by OCPD (*n* = 6), STPD (*n* = 3), AVPD and NRPD (*n* = 2 each), and PRPD and HIPD (*n* = 1 each). Controls were diagnosed with BPD and OCPD (*n* = 2 each) and PAPD and NRPD (*n* = 1 each). The group comparison by PD diagnosis can be seen in Table 2.

As for the PD traits assessed in comorbidity with each other, FM patients also differed significantly from non-FM subjects in the number of traits of particular PDs reaching diagnostical thresholds. AVPD, OCPD, PAPD, and BPD traits showed significantly higher frequency in FM patients (*p* < 0.05). HIPD and NRPD traits were of borderline significance (*p* = 0.05). DRPD, PRPD, STPD, SCPD, and ASPD trait frequencies did not differ significantly between groups (see Table 3).

The comparison between the mean scores of the SCID-II-PD traits, counted as numbers of identified traits per patient, showed significant differences between study groups for DPD, PAPD, PRPD, NCPD, and BPD traits (Table 4).

The binary logistic regression analyses showed that DPD, STPD, SCPD, and BPD traits could be significant predictors of FM diagnosis (Cox and Shell R^2^ = 0.305, Nagelkerke R^2^ = 0.415) (see Table 5).

## 4. Discussion

In our study, the proportion of PDs diagnosed in patients with FM was greater than that found in non-FM controls (56.7% vs. 18.2%), and this is consistent with other studies showing higher PD prevalence in FM compared to non-FM controls [20,26].

The PD frequency in our FM patients (56.7%) is in the middle of the interval for PD rates provided by other studies, i.e., the range from 8.7–13.5% [21,43] to 63.8–94.2% [20,25,27]. Our rate is almost the same as that in the study of Fu et al. (56%) [23] and relatively close to numbers provided by Sadr et al. (40.25%) [7] and Rose et al. (46.7%) [44].

As for particular PDs, our study shows that patients differed significantly from controls only in the frequency of BPD diagnosis that was more prevalent in FM (28.3% vs. 6.1%). This result is consistent with some studies that revealed BPD predominance over other PDs in FM [7,25,27,43]. In our study, BPD was followed by OCPD (11.3%) and STPD (5.7%), but the latter two PDs did not differ significantly from the controls in frequency.

However, when we compared not only PD diagnoses but PD traits as well, it was revealed that FM patients were characteristic by significantly higher frequency and/or severity of traits other than BPD-derived traits. Along with BPD, there was a significant difference for other cluster B PD traits, i.e., higher frequency and severity of PAPD traits and higher severity of NCPD traits.

In addition, our FM patients differed from controls in cluster C PD traits, i.e., higher frequency of AVPD and OCPD traits, and higher severity of DPD traits. High frequencies of AVPD and OCPD in FM have also been found in several studies, i.e., 10.7–61.9% and 23.3–71.1%, respectively [25,26,27,28]. AVPD alone was found to be the most frequent PD in the study of Fu et al. [23], and OCPD was found to be most frequent in the study of Rose et al. [44]. DPD was also found to be among the most frequent PDs in FM in the study of Thieme et al., 2004 [43].

Surprisingly, in our study, FM patients also had higher severity of PRPD traits that belong to cluster A. PRPD was found to be of high frequency only in the study of Fu et al. [23].

In our study, the only PD traits that did not differ significantly in frequency and/or severity were HIPD, ASPD, STPD, SCPD, and DRPD. Regarding HIPD traits, our data conflict with some studies that showed high rates of HIPD in FM [21,27]. This may be due to an overlap of HIPD traits with other cluster B PDs or due to some researchers’ preferences to overdiagnose HIPD when prominent dramatic traits are evident.

Similarly, ASPD was not detected as frequently in any of the PDs-in-FM studies that we analyzed. SCPD had been identified with relatively low frequency in FM (13.3–15.3%) [25,28]. STPD has not been reported yet as a frequent PD in FM either; Fu et al. [23] reported an STPD rate of only 12.5%. In our study, the STPD rate was also low (5.7% of STPD in our FM group), but STPD traits were relatively frequent (18.9%).

The inconsistencies regarding particular PD trait frequencies and severity in our study and in other studies may be explained by the different PD evaluation approaches (e.g., assessment of PD diagnosis vs. PD traits, PD trait frequency vs. severity, etc.). Thus, the results of the binary regression analysis that we performed to build a model of PD traits that may show the most impact in FM may address the issue. Traits that were found as predictors of FM diagnosis in our analysis (STPD, SCPD, BPD, DPD) belong to all three PD clusters: cluster A (STPD, SCPD), cluster B (BPD), cluster C (DPD). This may reflect the influence of eccentric, dramatic, and anxious PD clusters in combination as well as in FM subgroups. However, this hypothesis requires further research.

Two of the PDs (BPD and STPD) found to be FM predictors in our study belong to the “severe personality syndromes” defined by Millon as compared to other PDs [45,46]. DPD is also considered by some authors as a severe subtype of cluster C PDs [47]. These “severe” PDs may be hypothesized to be a reason for an extremely high FM comorbidity with other Axis I psychiatric disorders (affective, anxious, obsessive-compulsive, etc.), although this hypothesis also requires confirmation in further research. However, some indirect confirmations of a significant SPD impact in FM may be found in the study of Krupa et al. [48]. The researchers report that schizotypy along with other personality variables (depressive, irritable, and anxious temperaments, introversion, and neuroticism) is associated with resistance to treatment with serotonin and noradrenaline reuptake inhibitors (SNRIs) in fibromyalgia. Finally, there is a notion that SPD is not an “ordinary PD”, and it is even placed among schizophrenia spectrum disorders (e.g., “schizotypal disorder” but not “schizotypal personality disorder” in ICD-10 and ICD-11). This may lead to the exclusion of SPD subjects from PDs in FM research projects and cause an underestimation of the problem.

There are some limitations in our study. The frequency of some PDs was low due to the small sample size. Our results require replication in samples of a larger size. However, we believe that an approach based on PD trait analysis may be of value.

This study was held in a tertiary neurological setting, and this may have resulted in some selection bias of more severe FM and PD cases than those in the general population, primary medical care, or rheumatological settings. For this reason, our findings may have some limitations in generalizability and should be extrapolated to FM subjects from other settings with caution. There may also be another source of selection bias, as all patients gave their informed consent for and underwent a psychiatric consultation. This may partially explain the impact of severe PDs (BPD and STPD). However, this approach allowed us to go beyond a self-administered subjective PD assessment and to arrange a face-to-face psychiatric interview using a validated objective diagnostical tool (SCID-II). Thus, we believe that this could be considered a strength of our study and that our findings could be of value for consultation–liaison psychiatric/psychotherapeutic/psychological services in neurological settings.

## 5. Conclusions

Our results may reflect an association between FM and the personality traits of all three clusters (A, B, and C) in FM patients, possibly with the predominance of severe PD traits, particularly borderline, schizotypal, and dependent. However, the most consistent evidence seems to be for borderline PD.

## Figures and Tables

**Table 1 biomedicines-11-03162-t001:** Demographic, health status, and psychometric characteristics of main (FM patients) and control (non-FM subjects) groups of the study.

Variables	FM (*n* = 53)	Non-FM (*n* = 33)	*p*
Mean age, years (SD)	46.8 (14.6)	43.6 (12.4)	0.313 *
Female, *n* (%) Male, *n* (%)	47 (88.7) 6 (11.3)	24 (72.7) 9 (27.3)	0.058 **
Married or with partner, *n* (%) Unmarried or no partner, *n* (%)	29 (54.7) 24 (45.3)	14 (42.4) 19 (57.6)	0.268 **
Employed, *n* (%) Unemployed, *n* (%)	26 (49.1) 27 (50.9)	31 (93.9) 2 (6.1)	<0.001 **
Higher education, *n* (%) Basic education, *n* (%)	29 (54.7) 24 (45.3)	27 (93.9) 6 (6.1)	0.01 **
Disabled, *n* (%) Nondisabled, *n* (%)	11 (20.8) 42 (79.2)	1 (3.0) 32 (97.0)	0.021 **
EQ-5D VAS (±SD)	46.8 (±18.1)	82.2 (±16.6)	<0.001
BMI (±SD)	27.0 (±4.7)	23.4 (±3.3)	0.008 *
FIQR (±SD)	54.1 (±18.8)	2.1 (±3.5)	<0.001
FIRST (±SD)	4.8 (±1.3)	0.5 (±1.1)	<0.001
HADS-A (±SD)	10.9 (±4.5)	2.9 (±2.2)	<0.001
HADS-D (±SD)	7.9 (±3.4)	2.1 (±1.8)	<0.001
SOMS-2 (±SD)	29.3 (±9.0)	9.9 (±8.1)	<0.001 *

* Mann–Whitney U-test, ** Pearson’s χ2 test, EQ-5D VAS —EuroQol 5 Dimensions Health-related Quality of Life Questionnaire derived Visual Analog Scale, BMI—body mass index, FIQR—Revised Fibromyalgia Impact Questionnaire, FIRST—Fibromyalgia Rapid Screening Tool, HADS—Hospital Anxiety and Depression Scale (HADS-A—anxiety subscale, HADS-D—depression subscale), SOMS-2—Screening for Somatoform Symptoms Scale.

**Table 2 biomedicines-11-03162-t002:** Personality disorder SCID-II diagnoses in the main (FM patients) and control (non-FM subjects) groups of the study.

PDs *	FM (*n*, %), *n* = 53	Non-FM (*n*, %), *n* = 33	*p* **
AVPD	2 (3.8)	0 (0)	0.259
DPD	0 (0)	0 (0)	NA
OCPD	6 (11.3)	2 (6.1)	0.415
PAPD	0 (0)	1(3.0)	0.203
DRPD	0 (0)	0 (0)	NA
PRPD	1 (1.9)	0 (0)	0.428
STPD	3 (5.7)	0 (0)	0.165
SCPD	0 (0)	0 (0)	NA
HIPD	1 (1.9)	0 (0)	0.428
NRPD	2 (3.8)	1 (3.0)	0.856
BPD	15 (28.3)	2 (6.1)	0.012
ASPD	0 (0)	0 (0)	NA
Any PD	30 (56.7)	6 (18.2)	<0.001
No PD	23 (43.3)	27 (81.8)	

* PD types: AVPD—avoidant, DPD—dependent, OCPD—obsessive-compulsive, PAPD—passive-aggressive, DRPD—depressive, PRPD—paranoid, STPD—schizotypal, SCPD—schizoid, HIPD—histrionic, NRPD—narcissistic, BPD—borderline, ASPD—antisocial, ** Pearson’s χ2 test, NA—not applicable.

**Table 3 biomedicines-11-03162-t003:** Personality disorder SCID-II traits in the main (FM patients) and control (non-FM subjects) groups of the study.

PDs *	FM (*n*, %), *n* = 53 ***	Non-FM (*n*, %), *n* = 33 ***	*p* **
AVPD	19 (35.8)	5 (15.2)	0.037
DPD	10 (18.9)	0 (0)	0.08
OCPD	31 (58.5)	11 (33.3)	0.023
PAPD	16 (30.2)	2 (6.1)	0.007
DRPD	11 (20.8)	2 (6.1)	0.064
PRPD	15 (28.3)	4 (12.2)	0.079
STPD	10 (18.9)	3 (9.1)	0.218
SCPD	10 (18.9)	2 (6.1)	0.096
HIPD	9 (17.0)	1 (3.0)	0.05
NRPD	20 (37.8)	6 (18.2)	0.05
BPD	26 (49.1)	5 (15.1)	0.001
ASPD	4 (7.5)	2 (6.1)	0.792

* PD types: AVPD—avoidant, DPD—dependent, OCPD—obsessive-compulsive, PAPD—passive-aggressive, DRPD—depressive, PRPD—paranoid, STPD—schizotypal, SCPD—schizoid, HIPD—histrionic, NRPD—narcissistic, BPD—borderline, ASPD—antisocial, ** Pearson’s χ2 test, *** total PD trait numbers exceed the number of patients in a group due to comorbidity between PD traits reaching diagnostical thresholds in individual patients.

**Table 4 biomedicines-11-03162-t004:** Personality disorder SCID-II mean score comparison for traits in the main (FM patients) and control (Non-FM subjects) groups of the study.

PDs *	FM, Mean (SD) *n* = 53	Non-FM, Mean (SD) *n* = 33	*p* **
AVPD	1.92 (2.083)	1.58 (1.821)	0.644
DPD	2.40 (2.051)	1.06 (1.298)	0.002
OCPD	3.79 (1.945)	3.03 (1.591)	0.052
PAPD	2.21 (2.231)	1.06 (1.435)	0.030
DRPD	2.30 (2.145)	1.27 (1.506)	0.031
PRPD	2.30 (1.967)	1.39 (1.478)	0.043
STPD	2.30 (2.044)	1.76 (1.521)	0.326
SCPD	1.83 (1.503)	1.24 (1.347)	0.064
HIPD	1.81 (2.020)	1.18 (1.334)	0.263
NRPD	3.83 (2.701)	2.27 (2.066)	0.007
BPD	4.68 (3.167)	2.27 (2.295)	<0.001
ASPD	0.51 (1.049)	0.48 (1.064)	0.815

* PD types: AVPD—avoidant, DPD—dependent, OCPD—obsessive-compulsive, PAPD—passive-aggressive, DRPD—depressive, PRPD—paranoid, STPD—schizotypal, SCPD—schizoid, HIPD—histrionic, NRPD—narcissistic, BPD—borderline, ASPD—antisocial, ** Mann–Whitney U-test.

**Table 5 biomedicines-11-03162-t005:** Personality traits as independent variables predicting FM diagnosis in binary logistic regression model.

PDs *	B	S.E.	Wald	Df	Sig.	Exp (B)	95% C.I. for EXP (B) Lower	95% C.I. for EXP (B) Upper
AVPD	−0.210	0.196	10.148	1	0.284	0.811	0.552	10.190
DPD	00.652	0.242	70.226	1	0.007 **	10.919	10.193	30.086
OCPD	0.012	0.181	0.004	1	0.947	10.012	0.710	10.442
PAPD	0.305	0.275	10.231	1	0.267	10.356	0.792	20.324
DRPD	−0.392	0.285	10.895	1	0.169	0.676	0.387	10.181
PRPD	−0.038	0.221	0.030	1	0.862	0.962	0.624	10.485
STPD	−0.470	0.233	40.045	1	0.044 **	0.625	0.396	0.988
SCPD	0.584	0.255	50.231	1	0.022 **	10.793	10.087	20.959
HIPD	0.101	0.194	0.271	1	0.603	10.106	0.757	10.618
NRPD	0.146	0.160	0.826	1	0.363	10.157	0.845	10.584
BPD	0.336	0.164	40.176	1	0.041 **	10.399	10.014	10.930
ASPD	−0.460	0.342	10.810	1	0.179	0.631	0.323	10.234
Constant	−10.443	0.725	30.956	1	0.047	0.236		

* PD types: AVPD—avoidant, DPD—dependent, OCPD—obsessive-compulsive, PAPD—passive-aggressive, DRPD—depressive, PRPD—paranoid, STPD—schizotypal, SCPD—schizoid, HIPD—histrionic, NRPD—narcissistic, BPD—borderline, ASPD—antisocial. ** *p* < 0.05.

## Data Availability

The data presented in this study are available from the corresponding author on request.

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
