# Peer review of "Personality Disorders and Traits of ABC Clusters in Fibromyalgia in a Neurological Setting"

_biomedicines, 2023, doi:10.3390/biomedicines11123162_

Round 1
Reviewer 1 Report
Comments and Suggestions for Authors
Overall, this is an article that is of interest to the general medical community and particularly to those with specific focus on fibromyalgia and interface between fibromyalgia and personality disorders.
Introduction: Well written and discusses background. Needs correction of English in several instances: proper tenses, etc. References are appropriate.
Materials and Methods: Appropriately presented. Fundamentally sound. They also show major drawback of this manuscript/study; that being the small n. Furthermore, all patients were obtained at the Kozhevnikov Neurology Clinic of Sechenov University in Moscow, Russia. This small n will contribute to generalization error and inability to extrapolate these single center findings to the widespread global population. Was a normality test conducted to ensure that the subjects evaluated and the control groups were normally distributed? Please present more clinical characteristics of the subjects and controls so we better understand the differences between these two groups. All we have is the age comparisons. Were FIQR's and SIQRs done between the two groups for comparison? Do we have BMIs between the two groups to compare?
Results: Given the drawbacks, from above in terms of n's and lack of comparator values between the subjects and controls, it is difficult to fully assess the impact of Fibromyalgia alone vs. different physical characteristics as influences in the number of personality disorders that were seen. The results are otherwise reported fairly.
Discussion: Greater discussion of lack of study size, unknown physical characteristics, risk of generalization errors, needs to be discussed in this section.
Comments on the Quality of English LanguageOverall, this is an article that is of interest to the general medical community and particularly to those with specific focus on fibromyalgia and interface between fibromyalgia and personality disorders.
Introduction: Well written and discusses background. Needs correction of English in several instances: proper tenses, etc. References are appropriate.
Materials and Methods: Appropriately presented. Fundamentally sound. They also show major drawback of this manuscript/study; that being the small n. Furthermore, all patients were obtained at the Kozhevnikov Neurology Clinic of Sechenov University in Moscow, Russia. This small n will contribute to generalization error and inability to extrapolate these single center findings to the widespread global population. Was a normality test conducted to ensure that the subjects evaluated and the control groups were normally distributed? Please present more clinical characteristics of the subjects and controls so we better understand the differences between these two groups. All we have is the age comparisons. Were FIQR's and SIQRs done between the two groups for comparison? Do we have BMIs between the two groups to compare?
Results: Given the drawbacks, from above in terms of n's and lack of comparator values between the subjects and controls, it is difficult to fully assess the impact of Fibromyalgia alone vs. different physical characteristics as influences in the number of personality disorders that were seen. The results are otherwise reported fairly.
Discussion: Greater discussion of lack of study size, unknown physical characteristics, risk of generalization errors, needs to be discussed in this section.
Author Response
Dear Reviewer,
We are very grateful for the valuable and fair comments. We tried to address all the mentioned issues.
We agree with the remark about the small study sample and discuss it in the study limitations section. Please see lines 279-293. As for the generalization error, the remark is also fair. We do not pretend to generalize our results on all the FM subjects, particularly in general population or rheumatology as well. But we still think that our study sample may be of particular interest as it has been recruited among neurologically/psychiatrically clinically observed FM subjects. And that was a rather laborious job as required a face-to-face clinical interview and it was not just limited to self-administered psychometrical scale as it is performed in most PDs in general medicine studies. We hope that this issue somehow apologies our relatively small sample size.
We also admit that our sample may represent a set of patients with more severe mental issues, PDs as well, e.g. because those were patients who gave their informed consent to talk to a psychiatrist. Not all FM patients agree to. This may be also a source of a generalizability limitation. But simultaneously, this why our results may be of primary interest for consultation-liaison psychiatrists/psychotherapists/psychologists who work or consult in neurological services/settings. We have stressed this point in the study limitation section accordingly. We have also addressed the issue in the manuscript title and in the abstract. We added “…in a neurological setting” to highlight the particular contingent of our FM subjects.
Thank you for the remark about a normality test. We have added information about it (Shapiro-Wilk test) and an explanation about a reason for non-parametrical criterion use (Mann-Uitney U-test). Please see lines 160-162.
As for more characteristics of the subjects and controls, we have comparison for mean age, gender distribution, marital status and employment in the Materials and methods section. We have also added educational status and BMI comparisons as well. Mean scores for the Revised Fibromyalgia Impact Question-naire (FIQR) and the Fibromyalgia Rapid Screening Tool (FIRST) have been also added. We also provide comparison for screened anxiety, depression (the Hospital Anxiety and Depression Scale - HADS), and somatization (SOMS-2). Please see Table 1 in the Materials and Methods section (lines 94-114).
Finally, we addressed the need for correction of English and used the relevant service of the publisher. Please see the Certificate attached.
Best regards,
Dmitry Romanov et al.
Reviewer 2 Report
Comments and Suggestions for Authors
This manuscript aimed to seek potential mental disorders in women with fibromyalgia, and succeeded in its goal. The disease is prevalent and studying its causes and consequences is important, which are key points in favor of this manuscript. However, there are several key issues that are main weaknesses in this study and which require further work and analyses, as is listed below.
1. Fibromyalgia is a peripheral nervous system disease with characteristic impairments in nerve conductance that are measurable but are not even mentioned throughout this manuscript. This must be amended.
2. Peripheral and central nervous system activities are inter-active and need to be discussed as such, which is not done here.
3. Both men and women may develop fibromyalgia and men and women patients must both be studied as separate group, which awaits completion and is an essential element is any project focused on fibromyalgia.
4. Above all, it is crucial to avoid a situation where a study like this one creates a situation where women and men suffering from a physiological disease would be categorized as mental disease patients which my lead to mal treatment at several different levels.
Author Response
Dear Reviewer,
We are thankful for your remark about the fibromyalgia essence. We neither call fibromyalgia a physiological disease nor a mental disorder.
We added the corresponding passage in the Introduction section to avoid the misunderstanding, as we are far from the “body vs. soul” dichotomy. We share the view with other authors about conceptualization of fibromyalgia as a multifactorial disorder with neurophysiological, immunological and cognitive elements (the UK clinical guidelines by the Royal College of Physicians, 2022) and a biopsychosocial perspective for understanding and addressing patients with fibromyalgia (Turk, D.C.; Adams, L.M., 2016). We agree with the recent view that the precise cause of fibromyalgia remains unknown and the central issue is proposed to be abnormal pain processing within the nervous system. Please see the first paragraph in the introduction section (Lines 31-42). We diagnose fibromyalgia according to the official criteria (the American College of Rheumatology, 2016) and just study impact of comorbid personality disorders that remain greatly underinvestigated in fibromyalgia. However, personality disorders seem to be rather important for treatment response with SNRI. Please see the recent study by Krupa et al [49] that we added in the Discussion section of the manuscript revision (lines 269-274).
As for the reviewer’s remark about only females in our study sample, there might be a kind of misunderstanding. We have as females, as males in both of our study groups (in patients and in controls as well). Please see Table 1 in the Materials and Methods section. In the main group there are 53 patients with fibromyalgia (47 females and 6 males) and in the control group there are 33 subjects without fibromyalgia (24 females, 9 males).
Best regards,
Dmitry Romanov et al.
Reviewer 3 Report
Comments and Suggestions for Authors
This study aims to identify the exact cluster (A, B, C) PDs or traits that are more prominent in FM and may be predictors of FM diagnosis. They collected data from 86 subjects (53 with FM and 33 controls without FM) was analyzed in an observational, cross-sectional, comparative study. The assessment of categorical PDs and traits was performed independently with the Structured Clinical Interview for Personality Disorders (SCID-II). Binary logistic regression was used to determine FM predictors among PDs traits.
Compared with controls, FM patients had higher rate of PDs diagnoses (56.7 vs 18.2%, P<.001). However, the rate was significantly higher only for borderline PD diagnosis (28.3% vs 6.1%, P<.05). The binary logistic regression analysis showed that schizotypal and schizoid (cluster A), borderline (cluster B), and dependent (cluster C) personality traits may be significant predictors of fibromyalgia (Nagelkerke R2 = 0.415). Their results showed association of FM with 21 personality traits of all three PD clusters: A (eccentric), B (dramatic), C (anxious). However, the 22 most consistent evidence seems to be for borderline PD.
1) The overall writing has some formatting issues, like wording, spacing, and some redundancy. I suggest the authors check the grammar and avoid any typos. More importantly, the writing needs improvement for readers to understand more easily.
2) The method part is lack of details. More detailed descriptions are needed to explain the method.
3) I would recommend the authors to polish the figures with higher resolution and consistent font size, which makes it easier to recognize. In some figures, the font size is very small.
4) The results are not quite sufficient. More discussions on the result part are needed. Moreover, I would suggest the authors discuss relevant work (PMID: 35286152), which helps expand the scope of the study.
Comments on the Quality of English LanguageN/A
Author Response
Dear Reviewer,
Thank you for your remark about the manuscript editing. We have used the publishers English editing service to improve our text. Please see the Certificate attached.
Thank you for the remark about the method part and details. We have extended the Materials and Methods section accordingly.
Unfortunately, we could not make the figures with higher resolution and consistent font sizes as used the manuscript template provided by the publisher. Please see here: https://www.mdpi.com/files/word-templates/sensors-template.dot Anyway, we are very sorry for any inconvenience.
Thank you for the remark about the results sufficiency. We have extended the discussion section accordingly. However, we could not discuss the work suggested by the reviewer (PMID: 35286152). It is titled “Risk and Outcome of Breakthrough COVID-19 Infections in Vaccinated Patients With Cancer: Real-World Evidence From the National COVID Cohort Collaborative”. Unfortunately, it contains neither data about fibromyalgia nor data about personality disorders. Is the provided PMID (35286152) correct?
Best regards,
Dmitry Romanov et al.

Round 2
Reviewer 1 Report
Comments and Suggestions for Authors
Thank you for addressing prior critiques and emphasizing limitations on the study. The manuscript is interesting and will be a valuable addition to the literature.
Reviewer 2 Report
Comments and Suggestions for Authors
The article is acceptablefor publication in its present form